# Fluidic Oscillators, the Effect of Some Design Modifications

**Masoud Baghaei and Josep M. Bergada \***

Department of Fluid Mechanics, Universitat Politècnica de Catalunya, 08034 Barcelona, Spain;
masoud.baghaei@upc.edu
**\*** Correspondence: josep.m.bergada@upc.edu; Tel.: +34-937398771

**Abstract:** The number of applications where fluidic oscillators are expected to be used in the future, is raising sharply, then their ability of interacting with the boundary layer to modify forces on bluff bodies, enhancing heat transfer or decreasing noise generation, are just few of the applications where fluidic oscillators can be used. For each application a particular pulsating frequency and amplitude are required to minimize/maximize the variable under study, force, Nusselt number, etc. For a given range of Reynolds numbers, fluidic oscillators present a linear relationship between the output frequency and the incoming fluid flow, yet it appears the modification of the internal fluidic oscillator geometry may affect this relation. In the present paper and for a given fluidic oscillator, several performance parameters will be numerically evaluated as a function of different internal modifications via using 3D-CFD simulations. The paper is also evaluating the relation between the momentum applied to the mixing chamber incoming jet and the oscillator output characteristics. The evaluation is based on studying the output mass flow frequency and amplitude whenever several internal geometry parameters are modified. The geometry modifications considered were: the mixing chamber inlet and outlet widths, and the mixing chamber inlet and outlet wall inclination angles. The concept behind this paper is, to evaluate how much the fluidic oscillator internal dimensions affect the device main characteristics, and to analyze which parts of the oscillator produce a higher impact on the fluidic oscillator output characteristics. For the different internal modifications evaluated, special care is taken in studying the forces required to flip the jet. The entire study is performed for three different Reynolds numbers, 8711, 16034 and 32068. Among the conclusions reached it is to be highlighted that, for a given Reynolds number, modifying the internal shape affects the oscillation frequencies and amplitudes. Any oscillator internal modification generates a much relevant effect as Reynolds number increases. Under all conditions studied, it was observed the fluidic oscillator is pressure driven.

**Keywords:** fluidic oscillators design; Computational Fluid Dynamics (CFD); flow control

## 1. Introduction

The reduction or enhancement of the lift and drag forces on any bluff-body via modifying the boundary layer employing Active Flow Control (AFC), must be regarded as a novel technology. The use of pulsating flow in (AFC) applications, allows reducing the energy required to modify the boundary layer around a given bluff body. Zero Net Mass Flux Actuators (ZNMFA) and Fluidic Oscillators (FO), are two good candidates to generate pulsating jets, the later having the advantage of employing no moving parts, therefore increasing its reliability. Although there exist few canonical shapes on (FO), it is of major interest to investigate the (FO's) performance when modifying their internal dimensions, then oscillation amplitude and frequency are expected to change. The present paper aims to bring some light to this matter. One of the initial evaluations of the fluidic oscillators

performance when modifying its internal shape, was made in 2013 by Bobusch et al. [1], they made some suggestions regarding the mixing chamber inlet width in order to modify the fluidic actuator output frequency. Vatsa et al. [2] studied, using the lattice Boltzmann method and based on the solver PowerFLOW, two different configurations of sweeping jet fluidic oscillators (FO), which were further analyzed in 2015 by Ostermann et al. [3]. The two (FO) considered, resemble the ones studied by Bobusch et al. [1] and Aram et al. [4] respectively. Velocity profiles generated by the (FO) in quiescent air were compared with experimental data, results showed that the (FO) having sharp internal corners, similar to the one employed in [1], generated an output velocity distribution much more homogeneous than the oscillator having rounded internal corners. The results from the two different configurations were compared to identify similarities and differences between the designs, suggestions of how these differences may affect applications, were made.

Woszidlo et al. [5], studied the same configuration previously evaluated by Gartlein et al. [6]. Both configurations resemble the one studied by Bobbush et al. [1], the main differences resided in the output shape. In Woszidlo et al. [5] and Gartlein et al. [6], just a single output was considered. In this new paper, Woszidlo et al. [5], focused their attention in analyzing the flow phenomena inside the mixing chamber and the feedback channels. They also observed that, the increase of the mixing chamber inlet width was tending to increase the output frequency, and rounding the feedback channels would diminish the generation of the separation bubbles on these channels.

Slupski and Kara [7], studied using 2D-URANS with the software Fluent a range of feedback channel (FC) geometry parameters, the sweeping jet actuator configuration was the same as the one analyzed by Aram et al. [4]. The effects of varying the feedback channel height and width for different mass flow rates were studied. All the simulations were performed for a fully-turbulent compressible flow, using SST k-omega turbulence model. It was found that, oscillation frequencies increased with increasing feedback channel height, up to a certain point and then remained unaffected, however, frequencies decreased by further increasing the feedback channel width.

An experimental and numerical study of a fluidic oscillator which could generate a wide range of frequencies (50–300 Hz), was studied by Wang et al. [8]. Their study focused on the oscillation frequency response for different lengths of the feedback channels, 2D compressible simulations were performed using sonicFoam with k-epsilon as turbulence model. An inverse linear relation between frequency and the length of feedback loops was observed, frequency increased when decreasing the feedback channel length.

In 2018, the same configuration previously employed by [1], although now using a single exit, was numerically evaluated in 3D at Reynolds 30000 using the SST turbulent model by Pandley and Kim [9]. Two geometry parameters, the mixing chamber inlet and outlet widths were modified. They observed a significant effect of the flow structure and the feedback channel flow rate when modifying the inlet width, negligible effects were observed when modifying the outlet width. The output frequency and amplitude effects whenever the (FC) and the mixing chamber (MC) lengths were modified, was studied using a 2D numerical model by Seo et al. [10], the fluid was considered as incompressible, the Reynolds number employed was 5000. They observed that an increase of the feedback channel length generated no modifications on the output frequency, the same observation was previously obtained by [11], in both cases the flow was defined as incompressible, being the reason why the simulations could not provide the correct information. On the other hand, the increase of the mixing chamber length, generated a clear reduction on the actuator output frequency. They defined the length scale to be employed to properly non-dimensionalize the oscillation frequency.

The present paper is presenting a numerical evaluation of the same fluidic oscillator (FO) configuration experimentally evaluated in [1]. The effects on the stagnation pressure, net momentum acting onto the jet, output mass flow and mixing chamber incoming jet inclination angle, among other parameters, are analyzed for four different internal geometry modifications, the (MC) inlet and outlet widths and the (MC) inlet and outlet wall inclination angles. Three different Reynolds numbers are considered. The four geometry modifications chosen have a considerable impact on the flow inside

the (MC) and the (FC's). The (MC) inlet width decisively affects the reverse flow in the (FC's) and the Coanda effect in the (MC), the (MC) outlet width drastically modifies the pressure inside the (MC), the (MC) inlet inclination angles, affect the Coanda effect and the bubble volume inside the (MC), finally, the (MC) outlet inclined walls drastically change the stagnation pressure in these particular walls, changing as well the (FC) dynamic pressure and amplitude.

## 2. Fluidic Oscillator Main Characteristics

The central part of the (FO) considered in the present study is introduced in Figure 1, the four internal geometries modified, the mixing chamber (MC) inlet and outlet widths and angles, are clearly shown. Figure 1 also introduces the positive and negative directions taken for each geometry modification.

Regardless of the configuration evaluated, an orthogonal 3D mesh with 2,242,000 cells was used to evaluate the flow at Reynolds numbers up to 16,034, for the baseline case configuration the respective maximum x+, y+ and z+ were of 1.8, 4.7 and 1.2, very similar values were obtained for the rest of the configurations. A mesh with 5,933,900 cells was used to perform all simulations at Reynolds number 32,068. The maximum values of x+, y+ and z+ obtained with this mesh and for the baseline case at Reynolds number 32,068, were of 0.9, 1.2 and 0.7, respectively. The rest of the configurations evaluated at Reynolds number 32068 generated very similar x+, y+ and z+ values. Figure 2 presents the mesh used for the simulations, showing as well the entire computational domain. The mesh independence test performed to validate the meshes used in the present paper, was presented in the authors previous paper [12], where the fluidic oscillator baseline case was analyzed for a set of different Reynolds numbers, including the largest Reynolds numbers analyzed in the present manuscript.

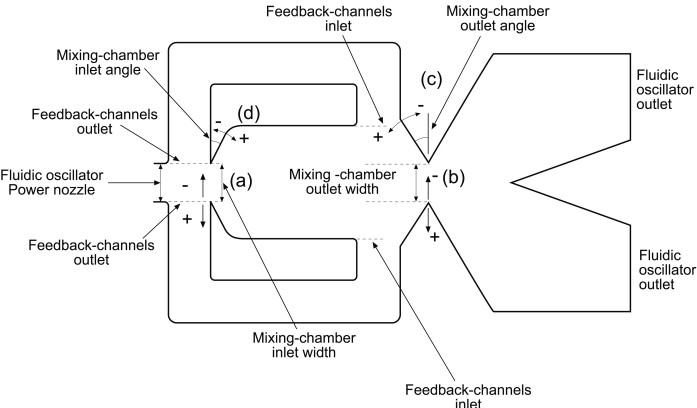

**Figure 1.** Fluidic oscillator mixing chamber internal dimensions modifications.

The boundary conditions employed in all simulations were, Dirichlet conditions for velocity and Neumann for pressure at the inlet. A relative pressure of $10^4$ Pa and Neumann conditions for velocity were considered at the two outlets. Dirichlet boundary conditions for velocity and Neumann for pressure were set to all walls.

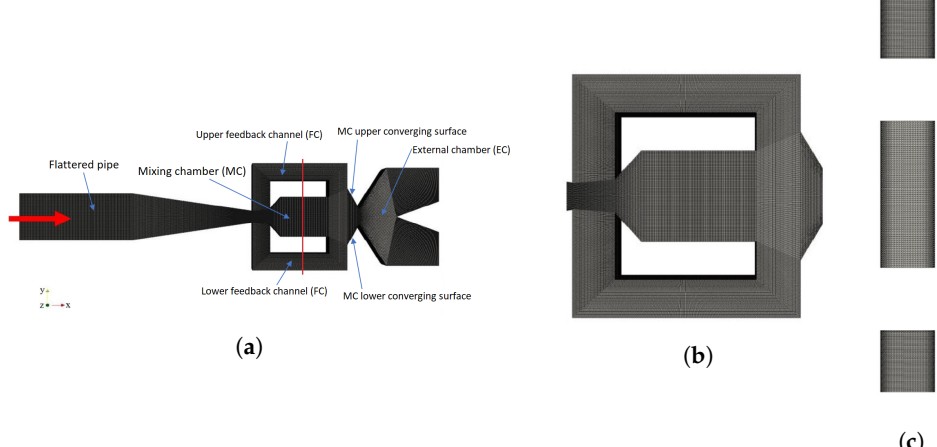

**Figure 2.** Grid used in the present study, (**a**) plane view, (**b**) zoom view, (**c**) side view of the mesh at the center of the mixing chamber, see the vertical line.

In the present study the flow was considered as turbulent, incompressible and isothermal, all simulations were three dimensional. The fluid used was water. Fluid dynamic viscosity was chosen to be $1.003 \times 10^{-3}$ Kg/(m s), the fluid density was 998.2 Kg/m$^3$. The turbulence model used was the DDES, which is a hybrid RANS-LES model, since according to the research undertaken by [4], this model generates a very close approach to the experimental results. The Spalart-Allmaras turbulent model was used for the RANS approach, while the Subgrid Scale (SGS) model was employed as the LES one. The parameter defining which turbulent model RANS-LES needs to be used is the distance between a given cell and the nearest wall, $d$. In the DES model, When $d < (C_{DES}\Delta)$, being $C_{DES} = 0.65$ a constant of the model and $\Delta$ the generic cell length, the RANS approach was used, and whenever $d > (C_{DES}\Delta)$ then the SGS model was employed. The DDES model is a modified version of the DES one, in which a new formulation with a filter function $f_d$ was introduced to avoid the so called grid-induced separation (GIS). In the DDES model, the switching mechanism between RANS and LES is not only dependent on the wall distance and grid spacing but also on the flow itself. The filter function $f_d$ is designed to take a value of 0 in the region where the boundary layer is attached, under this conditions the RANS model is used, and a value of 1 in the region where the flow is separated, under these conditions the switching criteria to employ whether the RANS or the LES model, is the one just defined for the DES model. The different constants involved in the turbulent models are, $c_{b1} = 0.1355$; $c_{b2} = 0.622$; $\sigma = \frac{2}{3}$; $\kappa = 0.41$; $c_{w1} = \frac{c_{b1}}{\kappa^2} + \frac{1+c_{b2}}{\sigma}$; $c_{w2} = 0.3$; $c_{w3} = 2$; $c_{v1} = 7.1$. Further information of the DDES model mathematical background employed along with the different constants involved is to be found in [13–17].

The software OpenFOAM was considered for all 3D simulations, finite volume method is the approach OpenFOAM uses to discretize Navier Stokes equations. Inlet turbulence intensity was set to 0.05% in all cases, Pressure Implicit with Splitting Operators (PISO), was used to solve the Navier Stokes equations, the time step being of $10^{-6}$ s, spatial discretization was set to second order.

The different velocities evaluated and defined at the inlet of the flattered pipe where the section was $10.3 \times 3.25 = 33.475$ mm$^2$, see Figure 2a, were 0.671 m/s, 1.2347 m/s, and 2.46 m/s, being the corresponding Reynolds numbers 8711, 16034 and 32068 respectively. The Reynolds numbers were based on the hydraulic diameter $D_h$ and the fluid velocity $V$ at the power nozzle, the same location was already used by [1]. One of the main characteristics of a fluidic oscillator is its linear output mass flow frequency behavior versus the inlet mass flow, usually represented as a function of the Reynolds number. The results obtained from the first four Reynolds numbers were used for comparison with the experimental results obtained by [1]. This comparison is presented in Table 1, further validating the 3D-CFD model introduced. Notice that in Table 1 the results obtained using two more Reynolds numbers 11152, 13593, are also presented. To compare the results, the Reynolds numbers presented in Table 1 were chosen to be exactly the ones employed by [1] in their experimental work. In fact

the code validation was previously presented in [12], see Figure 3 of this previously published paper. Another point which requires further discussion is the possible necessity of using a buffer zone. Also in the previously published paper [12] and for the baseline case at Reynolds number 16034, the fluidic oscillator outlet mass flow frequency obtained with and without a buffer zone was compared. The use of a buffer zone, involving an increase of over 9 million cells in the domain, gave an outlet frequency 1.4% smaller than the one obtained without a buffer zone. When considering the number of 3D cases to be studied in the present paper, and the small variation of the output frequency obtained when using a buffer zone, the authors decided not to implement the buffer zone in the present simulations.

**Table 1.** Comparison experimental and simulated results.

| Reynolds Number | 8711 | 11152 | 13593 | 16034 | 32068 |
|---|---|---|---|---|---|
| (CFD) Mass Flow Output Frequency (Hz) | 12.92 | 15.89 | 19.5 | 22.7 | 40.43 |
| Eperimental Output Frequency (Hz) [1] | 12.9 | 15.5 | 18.7 | 21.8 | - |
| Error in % | 0.15 | 2.5 | 4.2 | 4.1 | - |

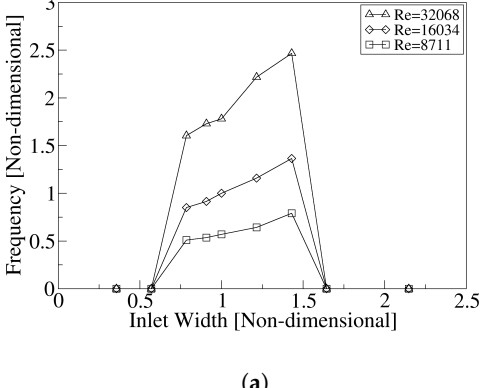

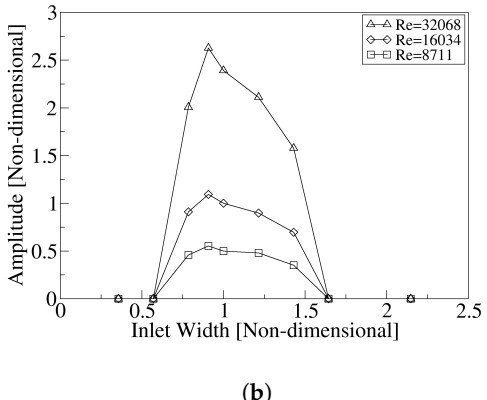

(**a**)  (**b**)

**Figure 3.** Fluidic oscillator output mass flow frequency (**a**) and peak to peak amplitude (**b**) as a function of the mixing chamber inlet width and for three different Reynolds numbers, 8711, 16034 and 32068.

To proceed with the non-dimensionalization and in order to generate graphs showing values around unity, the following dimensional parameters were employed. All dimensional parameters are based on values obtained for the baseline oscillator case at Reynolds number 16034. The (FO) outlet mass flow, was non-dimensionalised using the maximum value of the mass flow measured at one of the (FO) outlets. The maximum inclination angle of the main jet at the mixing chamber inlet, was used to non-dimensionalise the jet inclination angle at the mixing chamber inlet. The maximum momentum measured at one of the (FC) outlets, was employed to non-dimensionalise the momentum acting over the jet. The maximum value of the stagnation pressure measured at the mixing chamber outlet converging walls, was defined as the characteristic pressure for non dimensionalization. The characteristic length was chosen to be the oscillator's power nozzle hydraulic diameter $D_h$, as already employed in [1]. The fluid velocity at the (FO) power nozzle was employed as the dimensional characteristic velocity. The Reynolds number definition used to characterize the main flow, was: $Re = (\rho V D_h)/\mu$, where $\mu$ is the dynamic viscosity of the fluid. Time was maintained dimensional in all graphs.

## 3. Geometry Modifications Considered

The present study, is based on analyzing the effect of several geometry modifications on the (FO) outlet dynamic mass flow, frequency and amplitude. Four different modifications were evaluated, see Figure 1 and Table 2. Fluidic oscillator (MC) inlet width (a), was the first modification to be computed.

Eight different widths were analyzed, the maximum and minimum width ratio was respectively 2.14 and 0.35. Width ratio was defined as the generic inlet width divided by the original one. The inlet width was decreased by 64.4% and increased by 114.7%.

The (MC) outlet width (b), was respectively increased and decreased by a ratio of 1.82 and 0.17. Outlet width ratio was defined as the generic outlet width divided by the original one. A total of 8 different outlet width ratios were analyzed. The outlet width increase and decrease was of 82.3%. The (MC) outlet angle (c), was progressively increased and decreased versus its original value until reaching an outlet angular ratio respectively of 2 and 0.63, the definition of the outlet angular ratio is similar to the previous definitions already given. A total of 8 different outlet angular ratios were studied. The outlet angle maximum increase and decrease was respectively of 100% and 36.6%. Finally, the (MC) inlet internal angle (d) was as well modified, two different angles which increase versus the original one was respectively of 74.3%, and 93% were evaluated. It is interesting to mention that the inlet angles are directly linked with the position, shape an intensity of the Coanda vortices generated alternatively at both sides of the (MC). This angles also modify the shape and dimension of the (MC) bubble generated alternatively on both sides of the main jet, and according to [5,10,11] among others, there is a direct link between the (MC) bubble volume increase and the feedback channels mass flow. Table 2 summarizes all the different internal geometry modifications performed. All geometry modifications presented were evaluated for three different Reynolds numbers, 8711, 16034 and 32068.

**Table 2.** List of the different geometry ratios evaluated.

| Inlet Width/Reference Inlet Width | Outlet Width/Reference Outlet Width | Outlet Angle/Reference Outlet Angle | Inlet Angle/Reference Inlet Angle |
|---|---|---|---|
| 0.35 | 0.17 | 0.63 | 1 |
| 0.57 | 0.38 | 0.82 | 1.74 |
| 0.78 | 0.58 | 1 | 1.93 |
| 0.9 | 0.79 | 1.16 | |
| 1 | 1 | 1.31 | |
| 1.21 | 1.2 | 1.44 | |
| 1.42 | 1.41 | 1.56 | |
| 1.64 | 1.61 | 1.66 | |
| 2.14 | 1.82 | 2 | |

## 4. Momentum Acting on the Jet Entering the Mixing Chamber

In order to carefully evaluate the forces acting onto the main jet lateral surfaces, the momentum acting on any of the (FC) outlets is employed. The momentum is characterized by two terms, the (FC) mass flow and the static pressure at this particular location, Equation (1) defines each of the two terms.

$$M = \dot{m}_{out} * V_{out} + P_f * S_{out} = \dot{m}_{out}^2 / (S_{out} * \rho) + P_f * S_{out} \tag{1}$$

where, $\dot{m}_{out}, V_{out}, S_{out}$ and $P_f$, are respectively the instantaneous mass flow defined as $\dot{m} = \int_s \rho \vec{V} \vec{ds}$, the spatial averaged fluid velocity, the (FC) outlet surface and the pressure instantaneously appearing at any of the (FC) outlets, $\rho$ is the fluid density. The net momentum acting on the jet entering the (MC), is obtained when considering the forces defined by Equation (1) acting instantaneously on both (FC) outlets. The net momentum acting on the jet is composed by two terms, the net momentum due to the pressure term, which considers at each instant the pressure acting on both (FC) outlets, and the net momentum generated by the mass flow flowing along the (FC) and acting instantaneously on both (FC) outlets. At this point it is important to clarify that the momentum generated by the (FC) mass flow was obtained using the instantaneous mass flow to the power 2, divided by the section of the feedback channel and the fluid density, see the second term of Equation (1).

In the following sections and thanks to the simulations undertaken, it will be clarified, for the different geometry modifications considered, which is the role played by the net momentum due to the mass flow transferred across the (FC). This net momentum will be compared with the role undertaken by the net momentum due to the pressure difference acting onto the jet at the (FC) outlets. The role of the Coanda effect generated alternatively on both sides of the mixing chamber will also be investigated, in fact, this point will be particularly addressed when evaluating the effect of the (MC) inlet angle.

## 5. Results

### 5.1. Modifying the (Mc) Inlet Width

The first modification to be considered is the variation of the mixing chamber inlet width. Figure 3 introduces the variation of (FO) output mass flow peak to peak amplitude and frequency as a function of the different inlet widths evaluated. The first thing to notice is that whenever the inlet width falls below a minimum or is higher than a maximum value, the actual (FO) is not producing any outgoing frequency, see Figure 3a. The explanation why there is no flow oscillation when the actuator inlet width falls to a minimum, is based on the fact that, the mixing chamber incoming jet borders impinge onto the feedback channels (FC's) outlet internal vertical walls, creating a flow stream in the (FC's) which goes from left to right, from upstream to downstream, and along both (FC's) at the same time. At both (FC's) outlet internal vertical walls, a stagnation pressure point is generated, from this point, pressure waves are generated and sent from the feedback channels outlets to the inlets. The combination of these two effects prevents any feedback flow to move from downstream to upstream. On the other hand, when the mixing chamber inlet width is too large, a gap appears between the incoming jet and the mixing chamber inlet width borders. This small gap is enough to prevent a pressure increase at the (FC's) outlets, then it allows the fluid coming up from the (FC) inlet, to escape through this gap towards the mixing chamber.

In Figure 3b, is presented the effect of modifying the inlet width on the (FO) outlet mass flow peak to peak amplitude. As previously presented, for inlet widths exceeding a limit in any direction, whether too big or too small, the flow stops oscillating and the amplitude decays to zero. For the intermediate values it is seen that the amplitude is initially being highly affected by the inlet width, but as the width keeps increasing the amplitude decreases. It is also seen that the amplitude tendency is opposed to that of the frequency, small frequencies are linked with high oscillation amplitudes and vice versa. The reason why this is happening, it is clearly seen when applying the mass conservation equation between the (FO) inlet and outlets. For the present modification, when high widths are considered, the jet entering the mixing chamber (MC), suffers a relatively small oscillation inside the mixing chamber (MC), causing a small variation of amplitude at the oscillator exit. As the mixing chamber (MC) inlet width decreases, the jet oscillation amplitude inside the mixing chamber increases, the jet deflection angle at the mixing chamber inlet and outlet also increases and so does the fluidic oscillator output amplitude, see Figures 3–5. Based on the results presented in Figure 3, it can also be stated that the effects on output frequency and amplitude, are more relevant as the Reynolds number increases, but the trend already presented remains the same. The threshold at which the oscillation stops, appears to be rather independent of the Reynolds number.

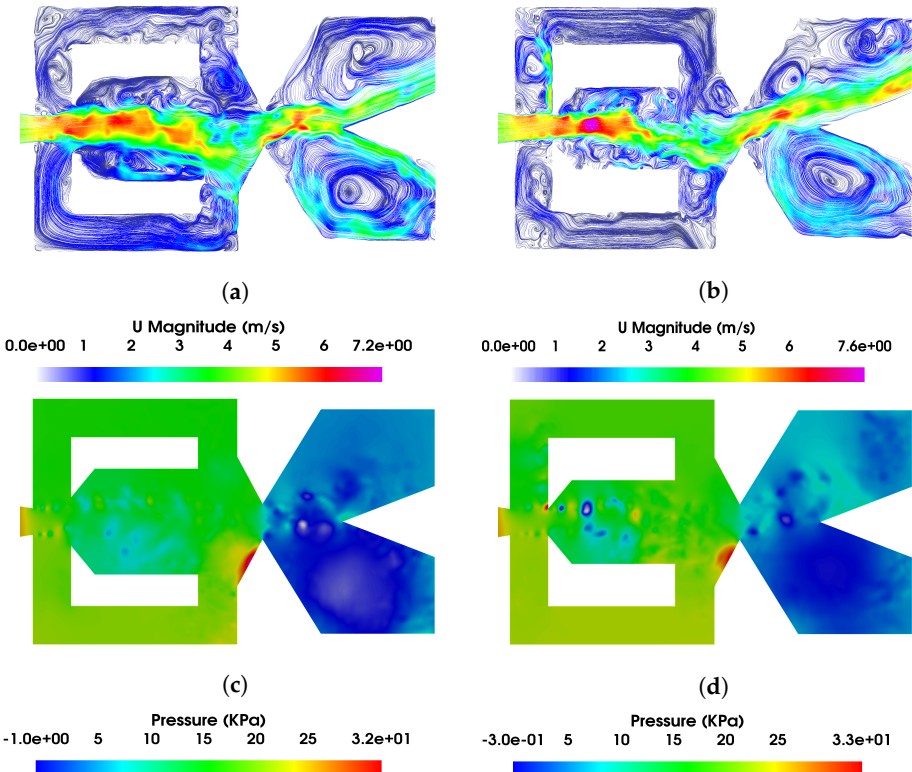

**Figure 4.** Fluidic oscillator internal velocity field (**a**,**b**), and pressure magnitude (**c**,**d**). Maximum inlet width (**a**,**c**), minimum inlet width (**b**,**d**). Reynolds number 16034.

Figure 4 introduces the flow field and pressure distribution inside the (FO), for the minimum and maximum (MC) inlet widths at which oscillation still appears, the Reynolds number is 16034. For the maximum inlet width, Figure 4a,c, the jet bending inside the (MC) is supported by the low pressure below the jet generated by the Coanda effect, see Figure 4c, at the external chamber the jet flows reattached to the surface of the wedge, generating two alternative vortices of nearly the same size on both sides of the external chamber. From Figure 5a, it is observed that no reverse flow appears at the (FO) upper outlet, see the curve characterizing the highest inlet width. Regarding the pressure distribution, the pressure is very much the same along the entire (MC), very small pressure fluctuations appear on the (FC's), alternatively pressurizing one (FC) or the other. The origin of the pressure waves, responsible of the (FC's) periodic pressurization, are the stagnation pressure points appearing alternatively at the (MC) outlet converging walls, see the red spots observed in Figure 4c,d.

At small inlet widths, Figure 4b,d, the (MC) incoming jet impinges alternatively at the (FC's) outlet internal vertical walls, generating an stagnation pressure point from which pressure waves are being sent alternatively from both (FC) outlets to the inlets, see the small red spot observed at the upper feedback channel outlet, Figure 4d. Reverse flow, from (FC) outlets to inlets is therefore generated, although for this particular case, the oscillation still exists. This is because the stagnation pressure at the (MC) converging walls, is acting over a surface about 20 times bigger than the one affected by the pressure at the (FC's) outlet internal vertical walls, and despite the fact the maximum pressure at the (MC) outlet converging walls, is for the present case, about 22% smaller than the maximum pressure existing at the (FC's) outlet internal walls, the time the stagnation pressure is acting on the (MC) converging walls, is 6.4 times longer than the time the stagnation pressure point appears at the (FC) outlet internal vertical walls, being this time difference along with the area the stagnation pressure acts, what maintains the oscillation. Although no figure related is presented in the present manuscript, if the inlet width would be further reduced, the stagnation pressure points at the (FC's) outlet vertical walls, would be appearing simultaneously on both vertical walls, generating reverse flow on both

feedback channels at the same time. Pressure waves would also be continuously transfered from the (FC's) upstream to downstream. Under these particular conditions oscillation would stop.

The evaluation of the dynamic values of the, net-momentum acting onto the jet entering the mixing chamber, the pressure at the (MC) outlet converging walls, the (MC) inlet inclination angle, the (FC's) mass flow and the (FO) output flow, greatly helps in understanding the (FO) dynamic performance. Figure 5 introduces the non-dimensional dynamic values of the mass flow through one of the (FO) outlets, the lower (FC) mass flow, the stagnation pressure at the (MC) outlet lower convergent wall, the net-momentum acting on the (MC) incoming jet and the (MC) inlet jet inclination angle. The Reynolds number was kept constant at 16,034. In each graph it is presented the non-dimensional dynamic value obtained for the baseline case, original actuator, and the corresponding ones characterizing the maximum and minimum inlet widths evaluated at which pulsating flow was observed. The first thing to notice is, that the dynamic stagnation pressure appearing alternatively at the (MC) converging walls, Figure 5c, is driving the oscillation, the rest of the graphs presented simply follow these pressure pulsations. From the oscillator upper outlet mass flow graph, Figure 5a, it is observed that the mass flow amplitude is directly linked with the reverse flow appearing at the (FO) outlets, the higher the reverse flow the higher the mass flow outlet amplitude, large reverse flows are associated to small inlet widths. This direct relationship is obvious when considering that the incoming (FO) mass flow is constant and given by the inlet boundary conditions, and at each instant, the mass flow through the two (FO) outlets must be the same as the inlet mass flow, the fluid is considered as incompressible. Therefore, if at some particular time, reverse flow appears at one of the (FO) outlets, the (FO) mass flow amplitude must increase to fulfill the continuity equation at each time instant.

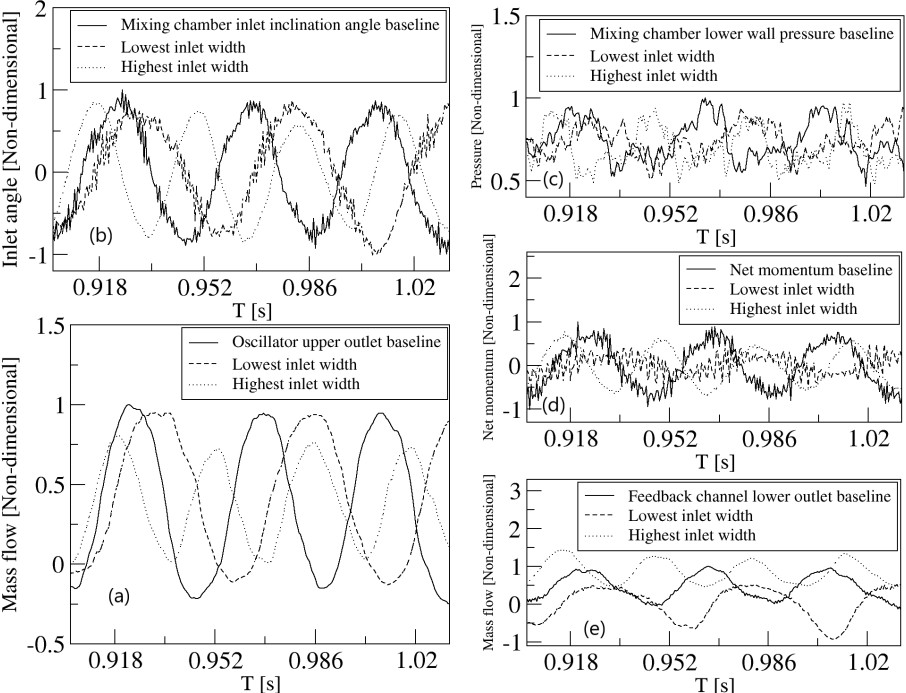

**Figure 5.** Dynamic effects of the (MC) inlet width modification on the main flow parameters, Reynolds number 16034. Each graph presents three non-dimensional curves characterizing results from the baseline, the lowest inlet width and the highest inlet width cases, and as a function of the dimensional time. In figure (**a**) the mass flow across the upper oscillator outlet is presented. Figure (**b**) introduces the temporal variation of the (MC) inlet inclination angle. Figure (**c**) presents the pressure at the (MC) lower inclined wall. The net momentum acting on the lateral sides of the main jet is presented in figure (**d**). Figure (**e**) characterizes the mass flow at the lower feedback channel outlet.

When observing the effects of the inlet width on the (FC's), Figure 5e, it is seen that, at small inlet widths the average mass flow is about zero, the flow inside the (FC's) is moving in both directions, yet the mass flow entering the (FC) outlet, reverse or negative flow, appears to be higher than the flow leaving such surface. If the inlet width would be further decreased, the reversed flow would keep increasing, eventually stopping the oscillation. As the inlet width increases, the mass flow inside the (FC's), although periodic, has a positive average value, meaning, there is a net mass flow moving from the feedback channels inlet to the outlet, at high inlet widths there is no reverse flow in the (FC's). The reason why at small inlet widths there is reverse flow in the (FC's), is the small stagnation pressure points generated alternatively at the (FC's) outlet internal vertical walls.

It is also relevant to highlight that, the peak to peak amplitude of the net momentum driving the jet oscillations inside the mixing chamber, is particularly small when the lowest inlet width is employed, Figure 5d, the jet is prone to fluctuate under these conditions, therefore a small pressure difference at the (FC's) outlets is sufficient to flip the jet. The jet oscillation amplitude inside the (MC) appears not to be affected by this fact, then the amplitude is higher than the one appearing at highest inlet widths. In other words, small inlet widths require small net momentums to flip the jet and the jet oscillation amplitude inside the mixing chamber is maximum. The (MC) incoming jet inclination angle, Figure 5b, suffers a reduction in the peak to peak amplitude of about 4.6% when comparing the maximum and minimum inlet widths, for the same conditions, the variation of the outlet mass flow amplitude, is about 16%. It appears the oscillation inside the (MC) is delimited by the (MC) upper and lower internal horizontal walls.

To properly understand the forces acting on the main jet lateral sides, the pressure and the mass flow terms of the net momentum acting on the main jet lateral sides are compared for the lowest, baseline and highest inlet widths, see Figure 6. Regardless of the inlet width studied, the pressure term of the net-momentum is much larger than the net-momentum mass flow term, indicating that the (FO) is pressure driven. Based on their observations and studying two different configurations of (FO), wu et al. [18] reached the same conclusion, although they could not bring a clear prove of it. From Figure 6, it is also important to realize that at low inlet widths, the pressure term of the net momentum is particularly scattered, as previously explained under these conditions in any of the feedback channels there is reverse flow and pressure waves travel in opposite directions.

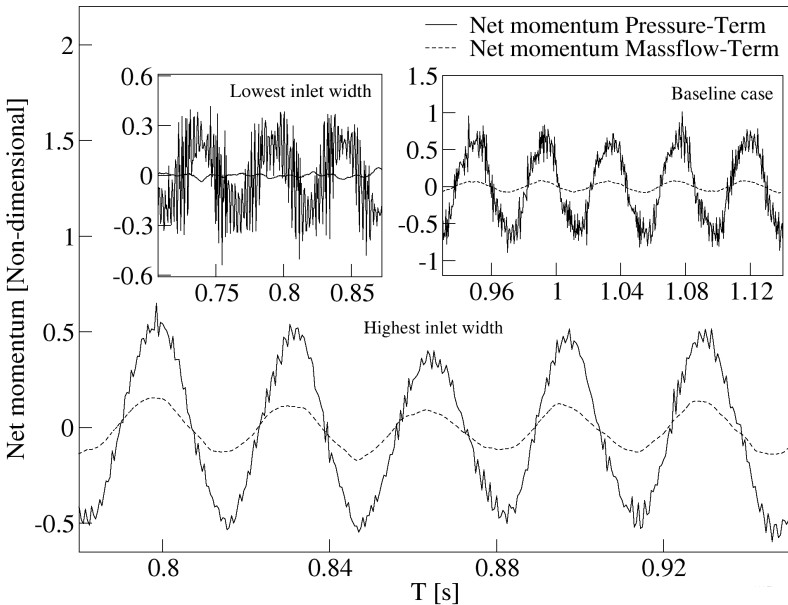

**Figure 6.** Comparison of the pressure and mass flow terms of the net-momentum and for three characteristic inlet widths, Reynolds number 16034.

From the dimensional temporal pressure plots at the (FC's) outlets, not presented in Figure 5, and when comparing the highest and smallest inlet widths studied, it was observed that the average static pressure at both (FC) outlets, decreased about 3000 Pa when the highest inlet width was used. Regarding the pressure fluctuation at both (FC) outlets, when the lowest inlet width was employed, the pressure was at any time, almost identical on both feedback channel outlets. While when the inlet width was maximum, the maximum pressure difference between the two (FC) outlets increased to about 2000 Pa. From the results obtained, it can be stated that at higher inlets widths, a higher net momentum onto the mixing chamber incoming jet lateral sides is required to bend the jet towards the opposite direction and therefore generate flapping. This phenomenon is understood when realizing that at small inlet widths, the Coanda effect helps in generating the required pressure difference inside the mixing chamber, to flip the incoming jet.

As Reynolds number increases to 32,064, there is an increase of the average pressure across the entire (MC) and therefore at the (FC's) outlets. The peak to peak amplitude of the mass flow inside the feedback channels, also increases. This is because the stagnation pressure is likely to increase with the velocity increase $P_0 = \rho V^2 / 2$. At Reynolds number 32,064 and for the lowest inlet width evaluated, the feedback channel mass flow reaches higher negative values than at Reynolds 16,034, but its average value remains quite constant and close to zero. In other words, at high Reynolds numbers, a higher net momentum acting on the (MC) incoming jet is required, to produce the jet flapping. The effect of Reynolds number on all (FO) dynamic parameters studied is presented in the last section of the paper.

### 5.2. Modifying (Mc) Outlet Width

The second dimensional evaluation undertaken consisted in analyzing the effect of modifying the mixing chamber outlet width. Figure 7, introduces the results obtained for the three Reynolds numbers studied. The first thing to observe, is that, an outlet width increase involves a reduction of the (FO) output frequency and amplitude, such reduction is more relevant as the Reynolds number increases. For example, at Reynolds 8711 and when comparing the values of the maximum and minimum outlet widths evaluated, the maximum decrease in frequency and amplitude was of about 7% and 60% respectively. At Reynolds 32,068, the respective decrease of frequency and amplitude was about 9% and 74%. In other words, the range of frequencies and amplitudes a given fluidic oscillator can produce, when modifying the outlet width, increases with the Reynolds number increase. The variation of the outlet width affects mostly the (FO) outlet amplitude. A point to consider when comparing Figures 3 and 7, is that the increase of the inlet width was bringing an increase in frequency and a decrease in amplitude, while an increase of outlet width generates a decrease in outlet frequency and amplitude. This opposite effect needs to be understood when evaluating the velocity fields under these four extreme conditions, these instantaneous velocity and pressure fields are presented in Figure 8.

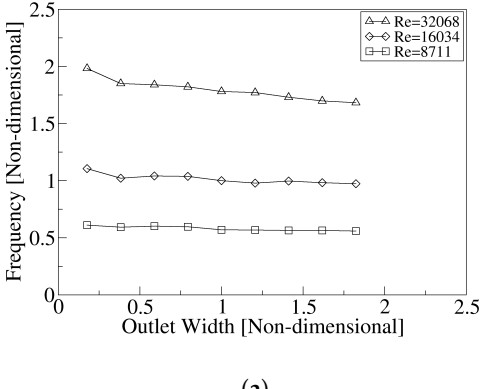
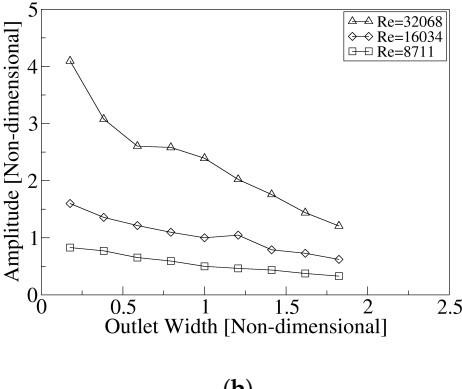

(**a**)                                 (**b**)

**Figure 7.** Fluidic oscillator output mass flow frequency (**a**) and peak to peak amplitude (**b**) as a function of the mixing chamber outlet width and for three different Reynolds numbers, 8711, 16034 and 32068.

Figure 8a,b, show the velocity vectors magnitude inside the oscillator for the highest and lowest outlet widths evaluated. Notice that when the (MC) outlet width is minimum, the velocity of the jet leaving the external chamber is maximum, more than four times the maximum velocity found for the rest of the cases studied in this paper. The average pressure in the (MC) is about nineteen times higher than for the rest of the cases studied. The fluid is pressurized due to the restriction effect caused by the small outlet width. The fluid stiffness in the (MC) and therefore its dynamic response is particularly high, explaining why high frequencies are linked to small outlet widths. At the (EC) the pressure is particularly low, see Figure 8d, in fact the relative negative pressure is of about 120 KPa, which is physically not possible and in reality shows that for small outlet widths cavitation is likely to appear at the (EC). Under these conditions, at the (FO) outlets, the section used by the flow to leave the oscillator is being reduced, leaving a large part of the outlet section in which reverse flow exists, huge spatial velocity differences are to be seen at the (FO) outlets. As a result, high (FO) outlet mass flow amplitudes are expected.

Figure 8c shows, for the highest outlet width studied, there is a low pressure area generated below the jet due to the Coanda effect, it can also be seen that the (FC) located below the (MC) is beginning to be pressurized, a clear stagnation pressure point is observed at the (MC) outlet lower converging wall. The mass flow spatial distribution at the (FO) outlets, appears to be much uniform, the (FO) outlets have a smaller surface through which reverse flow exists, being this directly linked to smaller outlet amplitudes. The (MC) is slightly pressurized, the fluid stiffness is low and so it is the (FO) outlet mass flow oscillating frequency. For the present (FO) and regardless of the dimensional modification, the maximum mass flow amplitudes at the (FO) outlets, are directly linked with the reverse flow existing at the outlets, the bigger the reverse flow the higher the (FO) outlet oscillation amplitude. The outlet width can be effectively used to control the outlet oscillations amplitude.

Regardless of the outlet width, a vortex at the external chamber upper side, can be spotted, yet its intensity and turning speed associated are much smaller for the highest than for the lowest outlet widths studied. The turning speed of this particular vortex for the lowest and highest outlet widths studied, was respectively of 301 and 59 rad/s, which explains why the pressure at the (EC) is particularly low at small outlet widths.

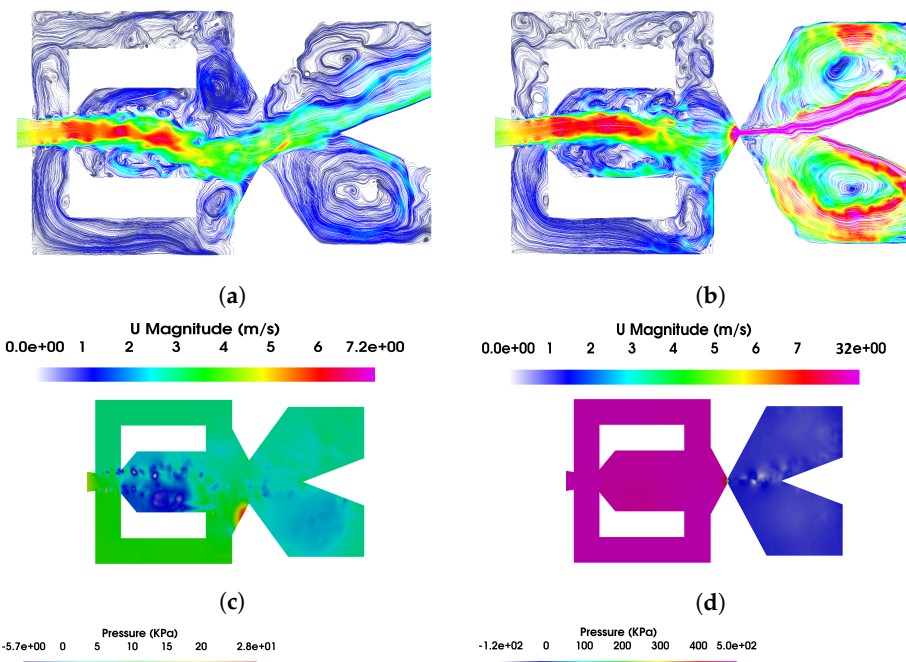

**Figure 8.** Fluidic oscillator internal velocity field (**a**,**b**) and pressure magnitude (**c**,**d**). Maximum outlet width (**a**,**c**), minimum outlet width (**b**,**d**). Reynolds number 16034.

The same dynamic parameters evaluated in the previous section and introduced in Figure 5, are now being presented in Figure 9 for the maximum, minimum and baseline (MC) outlet widths studied. When the highest (MC) outlet width is employed, the (FO) outlet mass flow amplitude is minimum, there is no reverse flow at any time. As the (MC) outlet width decreases, the reverse flow at the (FO) outlets keeps increasing, consequently the (FO) mass flow amplitude increases, see Figure 9a. Another relevant effect associated to the decrease of the outlet width, is the progressive increase of the (MC) pressure, notice that the average value of the stagnation pressure fluctuations presented in Figure 9c, increases around 19 times when comparing the maximum and minimum outlet width values. For the lowest outlet width evaluated, the stagnation pressure fluctuations at the (MC) lower converging wall, show a quasi-chaotic behaviour, probably due to the stagnation pressure points appearing simultaneously on both (MC) converging walls, see Figures 8d and 9c, although clear oscillation signs are still to be seen. Under these conditions, the curve representing the net momentum is particularly scattered. This fact is clearly understood when considering that the main term of the net momentum is the pressure term, as observed in Figure 6 for the previous case. Regarding the net momentum responsible of the jet fluctuations inside the (MC), see Figure 9d, its amplitude is about 38% smaller for high outlet widths than for the smaller ones. Notice as well that for the highest outlet width, the stagnation pressure peak to peak amplitude is about 40% smaller than for the smallest outlet width evaluated. These values suggest a direct relation between the stagnation pressure peak to peak amplitude and the net momentum driving the jet oscillations.

Another point to be discussed may be, why the net momentum shows rather a sinusoidal curve when the stagnation pressure oscillations, specially at the lowest outlet width, looks rather chaotic. The reason is the integration effect the (FC's) outlets are having on the pressure oscillations. The study of the (MC) incoming jet oscillation angle, see Figure 9b, shows that the smallest oscillation amplitude appears when the highest outlet width is employed. The maximum jet oscillation angle amplitude, obtained for the minimum outlet width, is about 3% higher than the one generated for the baseline case, and about 29% higher than the one obtained when using the maximum outlet width. From Figure 9, a direct correlation between the peak to peak stagnation pressure, the (MC) inlet angle amplitude, the (FO) output and (FC) mass flows amplitude, as well as the net momentum amplitude, appears to exist. Small stagnation pressure amplitudes generates small amplitudes in all these parameters.

When evaluating the feedback channel mass flow, Figure 9e, regardless of the outlet width, it is observed there is an average mass flow flowing from both feedback channel inlets to the outlets. Its average value remains pretty much constant regardless of the outlet width chosen, although it was observed it increased with the Reynolds number increase. For the smallest (MC) outlet width evaluated, just a very small reverse mass flow exists on both feedback channel outlets, the reverse flow increases with the Reynolds number increase. (FC) reverse flow is associated to the alternative appearance of the stagnation pressure points at the (FC) outlets internal vertical walls, as it was observed in Figures 4d and 5e for the smallest inlet width. At high mixing chamber outlet widths, there is no (FC) reverse flow. Regarding the (FC) mass flow amplitude, a decrease of about 32% is observed, when comparing the values of the highest outlet width with the ones obtained at the lowest outlet width. The decrease in feedback channel mass flow amplitude as outlet width increases can be more clearly seen at high Reynolds numbers, in general it can be said that all differences can be seen more clearly at high Reynolds numbers, the dynamic values at several Reynolds numbers are not presented in the present paper. The phase lag between the oscillator output mass flow and the feedback channel mass flow is about $4.4 \times 10^{-3}$ and $2.4 \times 10^{-3}$ s, respectively for the lowest and highest outlet widths at Reynolds 16034. It seems high phase lag values are related to high (FO) output mass flow amplitudes.

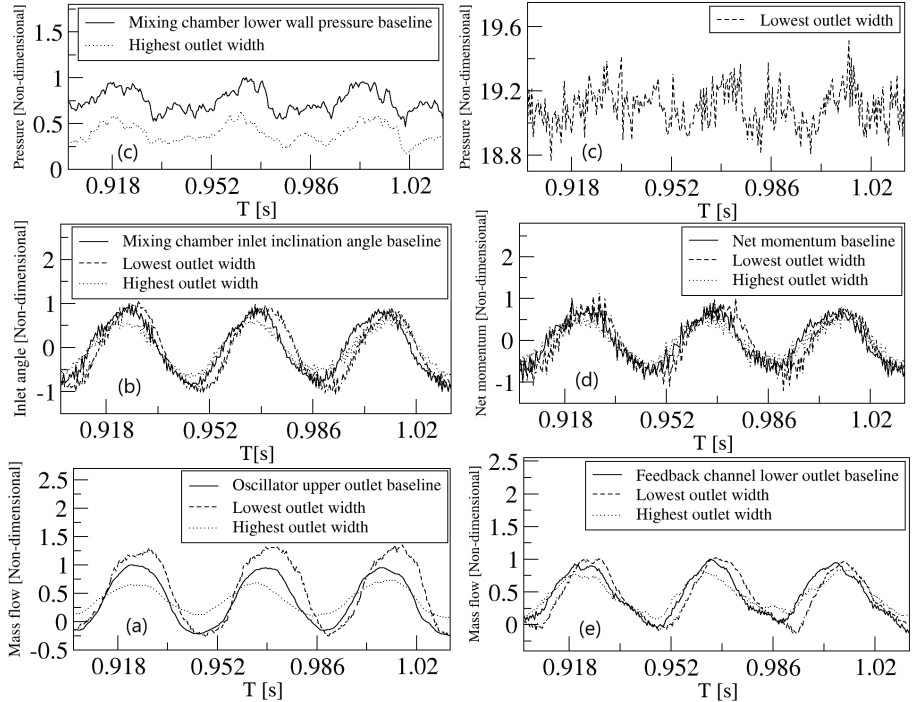

**Figure 9.** Dynamic effects of the (MC) outlet width modification on the main flow parameters, Reynolds number 16034. Each graph presents three non-dimensional curves characterizing results from the baseline, the lowest and highest outlet width cases, and as a function of the dimensional time. In figure (**a**) the mass flow across the upper oscillator outlet is presented. Figure (**b**) introduces the temporal variation of the (MC) inlet inclination angle. Figures (**c**) presents the pressure at the (MC) lower inclined wall. The net momentum acting on the lateral sides of the main jet is presented in figure (**d**). Figure (**e**) characterizes the mass flow at the lower feedback channel outlet.

Although not directly presented in the present manuscript, at low (MC) outlet widths, the pressure difference between the upper and lower (FC) channel outlets, was particularly high, a maximum pressure difference of 4200 Pa was measured. For the highest outlet width studied, this maximum pressure difference was of 2500 Pa. The peak to peak stagnation pressure oscillations amplitude at the (MC) converging walls, was specially high at low outlet widths and high Reynolds numbers. From Figure 9b,d, it is observed that for the lowest outlet width studied, the peak to peak net momentum amplitude acting onto the jet entering the (MC), and the peak to peak inlet angle amplitude of the jet at the same point, are respectively of 43% and 30% higher than the respective values obtained when evaluating the highest outlet width. A final relevant point to highlight on the average pressure, is that for most of the cases presented in this paper, and regardless of the modification considered, as Reynolds number increases from 16034 to 32068, the average pressure at the (MC) outlet converging walls, increased by approximately 150%. Yet there are two exceptions, one of them is whenever the highest (MC) outlet width is employed, for this particular case, the same increase of Reynolds number brings an increase of the average pressure of about 12.7%. The second exception appears when the lowest (MC) outlet width is employed, being the increase of average pressure of 316% for the same increase of the Reynolds number. For highest outlet widths, the central core of the main jet flows towards the (EC) without impinging on the (MC) converging walls, just some fluid particles located at the lateral sides of the jet impinge on the (MC) converging walls, therefore explaining why the average spatial pressure on these walls is particularly small. Regardless of the geometry modification evaluated, the stagnation pressure peak to peak amplitude at the (MC) converging walls, increases with the Reynolds number increase. The detailed evaluation of this particular parameter is to be found in the final part of this paper.

### 5.3. Modifying the (Mc) Outlet Angle

Figure 10 presents the results obtained when modifying the (MC) outlet angle and for the three Reynolds numbers evaluated. The (FO) output mass flow frequency and amplitude keeps decreasing as the inclination angle increases. The frequency effect is perfectly understandable once it is realized that the (MC) outlet converging walls, play a key role regarding the flow directed towards the (FC's) and the pressure waves transmission. Not only the position of the stagnation pressure point is modified by this angle but also its magnitude will be affected. The stagnation pressure maximum value as well as the peak to peak pressure amplitude, was observed to decrease with the (MC) angle increase. High frequencies are linked to high stagnation pressure values and vice versa. Furthermore, high (MC) outlet angles tend to direct the pressure waves towards the (FC) located opposite to the wall where the main jet impinges, the main flow stream is directed towards the (FO) outlet. On the other hand, small (MC) outlet angles, have associated a much wider area where stagnation pressure exists, maximum stagnation pressure values and peak to peak amplitudes are obtained under these conditions. Small angles direct the mass flow and specially the pressure waves towards the (FC) located next to the wall the main jet impinges, periodically pressurizing the (FC), compare Figure 11a,c with Figure 11b,d. This is the reason why, small (MC) outlet angles, have associated higher frequencies and amplitudes.

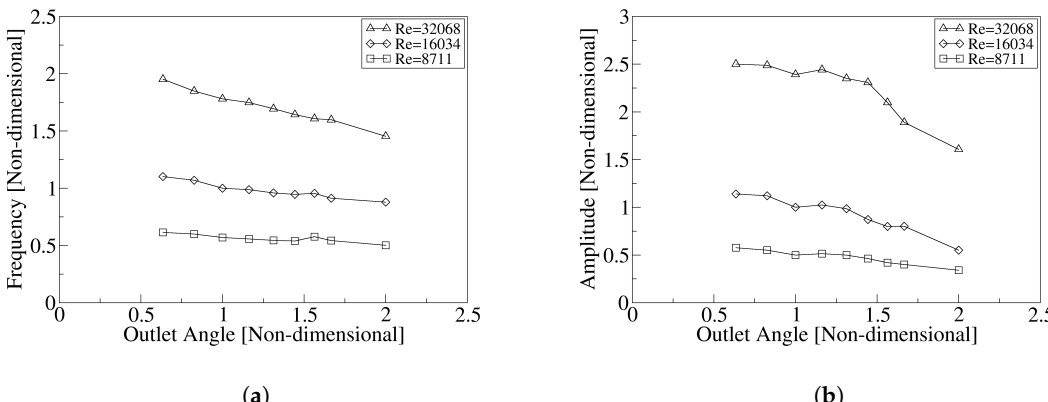

(**a**)                                    (**b**)

**Figure 10.** Frequency (**a**) and amplitude (**b**) of the (FO) outlet dynamic mass flow, as a function of the mixing chamber outlet angle and for three different Reynolds numbers, 8711, 16034 and 32068.

The effect of the (MC) outlet angle on the (FO) outlet mass flow amplitude, can be further explained when taking into account that, higher angles tend to direct the main flow towards the (FO) outlet central horizontal axis, therefore tending to decrease the jet deflection. Notice from Figure 12 that the amplitude of all parameters is particularly small at high outlet angles. The flow leaves the (EC) rather tangentially to the wedge walls and across one of the outlets at a time, generating a large vortex on the opposite outlet of the (EC). For high (MC) outlet angles, the fluid velocity at the (FO) outlets, is rather uniform across one of the exits at a time and always leaves the oscillator, there is no reverse flow, see Figure 12a. As the angle decreases, the outlet maximum fluid velocity magnitude increases, for a given oscillation period and during approximately one third of the period, the mass flow in any of the two exits enters the oscillator, for the rest of the period the fluid leaves the oscillator at a relatively high speed, the peak to peak output mass flow amplitude is maximum under these conditions, see Figure 12a. Again we are observing that large (FO) output mass flow amplitudes, have associated reverse flow at the (FO) outlets.

Figure 11 represents the oscillator overall velocity and pressure fields when the outlet angles are respectively the largest and smallest studied. From Figure 11b, which characterizes the smallest angle evaluated, it is noticed that the jet impinges nearly perpendicular to the (MC) lower converging wall, generating a large area where the stagnation pressure acts, see Figure 11d. The stagnation pressure and its peak to peak amplitude, reach their respective maximums under these conditions, see

Figure 12c. Pressure waves and some fluid flow are directed from the lower (FC) inlet to the outlet. the lower feedback channel is pressurized, Figure 11d. On the other hand, whenever the output angle increases, Figure 11a, the jet leaving the (MC), tends to run tangential to the (MC) converging walls, therefore directing a smaller amount of fluid through the feedback channels, and even more important, the formation of a high pressure stagnation point is reduced to a very small converging walls section, its peak to peak amplitude and maximum value are also minimized, see Figures 11c and 12c. As a result the time needed for the main jet to flip over increases, and accordingly the oscillation frequency decreases. It is also interesting to realize that at the external chamber (EC), a large vortex is generated at the opposite exit from the one the flow is leaving the amplifier. In fact the vortex covers the entire opposite exit, preventing flow from outside the (FO) to enter into the (EC). This vortex intensity was observed to be slightly higher as the angle decreased. For the smallest (MC) outlet angle, the turning speed associated to this particular vortex was of 117 rad/s, while when highest outlet angle is used, the turning speed was of 67 rad/s.

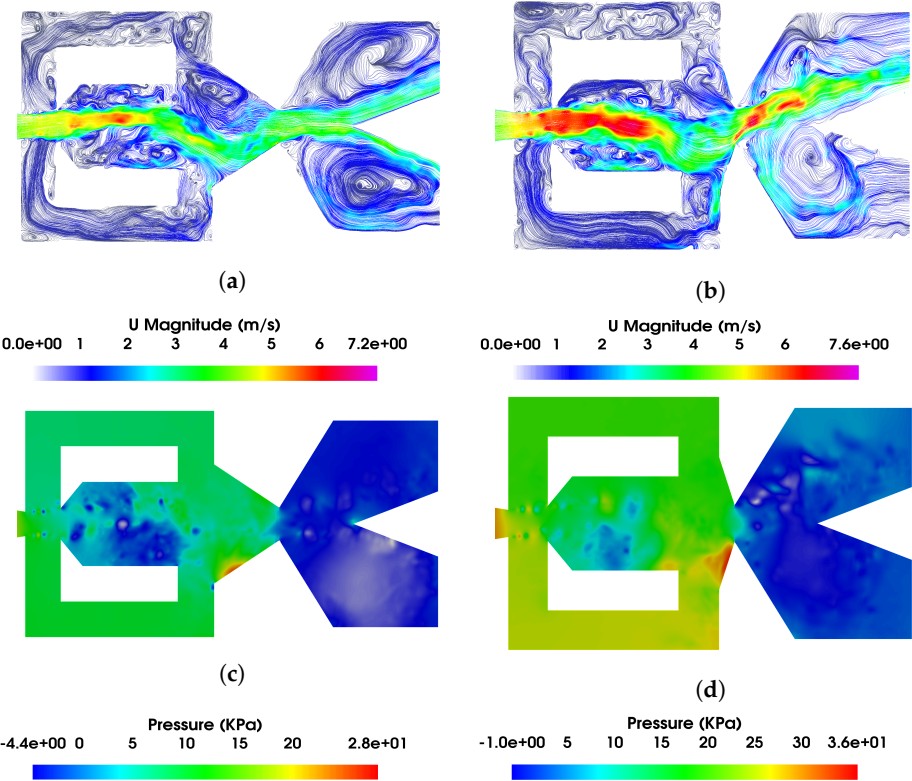

**Figure 11.** Fluidic oscillator internal velocity field (**a**,**b**) and pressure magnitude (**c**,**d**). Maximum outlet angle (**a**,**c**), minimum outlet angle (**b**,**d**). Reynolds number 16034.

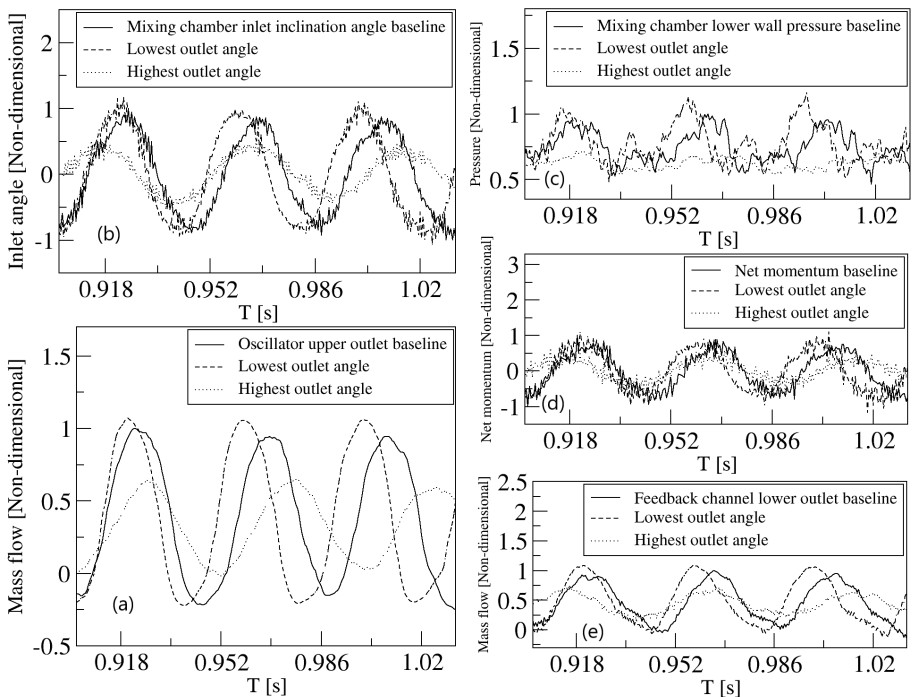

**Figure 12.** Dynamic effects of the (MC) outlet angle modification on the main flow parameters, Reynolds number 16034. Each graph presents three non-dimensional curves characterizing results from the baseline, the lowest and highest outlet angle cases, and as a function of the dimensional time. In figure (**a**) the mass flow across the upper oscillator outlet is presented. Figure (**b**) introduces the temporal variation of the (MC) inlet inclination angle. Figures (**c**) presents the pressure at the (MC) lower inclined wall. The net momentum acting on the lateral sides of the main jet is presented in figure (**d**). Figure (**e**) characterizes the mass flow at the lower feedback channel outlet.

The effects on the (FO) outlet mass flow, the (FC) mass flow, the net momentum acting on the (FC) outlets, the stagnation pressure on the (MC) converging walls, and the (MC) jet oscillation angle, for the baseline case, the lowest and highest mixing chamber outlet angles evaluated, at Reynolds number 16034, are presented in Figure 12. The first thing to observe is that the (MC) outlet angle modification, generates a clear effect on the (FO) main parameters. As the outlet angle increases, a clear peak to peak amplitude reduction of all measured parameters was observed. For the (MC) smallest outlet angle studied, the mass flow frequency and amplitude at the (FO) outlet are maximum, the reverse flow is also the largest. The peak to peak mass flow amplitude at the (FC's), is about 53% smaller when using the highest (MC) outlet angle than when using the smallest one. Under all (MC) outlet angles studied, the (FC) mass flow always goes from the (FC) inlets to outlets, for the lowest outlet angle, the minimum (FC) mass flow is about zero, see Figure 12e. The average pressure at the (MC) lower converging wall, is around 22% lower for the highest (MC) outlet angle than for the lowest one. When the lowest (MC) outlet angle was employed, a clear difference between the static pressure at the (FC) outlets was observed, a maximum pressure difference between both outlets of over 3000 Pa was measured. For the lowest (MC) outlet angle, the net momentum acting onto the lateral sides of the main jet entering the (MC), is about 38% higher than when the highest angle is used, Figure 12d. This effect helps in generating a much larger jet oscillation amplitude in the (MC) and at the (FO) outlet mass flow, see Figure 12a,b. The conclusion is that high oscillation amplitudes are linked with high pressure variations on the (MC) converging walls and therefore on the (FC) outlets.

## 5.4. Modifying the (Mc) Inlet Angle

The flow effects caused by the modification of the (MC) inlet angle is presented in this section, just three angles including the baseline case, were considered, see Figure 13. It was observed that,

the (FO) outlet mass flow peak to peak oscillation amplitude, when compared with the baseline case, increased an 8.9% for an inlet angle increase of 74%. When the inlet angle increased by 93%, the (FO) outlet mass flow amplitude increased versus the baseline case, just 1.6%. Regarding the (FO) outlet mass flow oscillating frequency, it increases very slightly with the inlet angle increase, in fact, as it has been introduced in all previous cases, the frequency and amplitude variations are more relevant as the Reynolds number increases. Figure 13 also clarifies that the same trend is observed regardless of the Reynolds number considered.

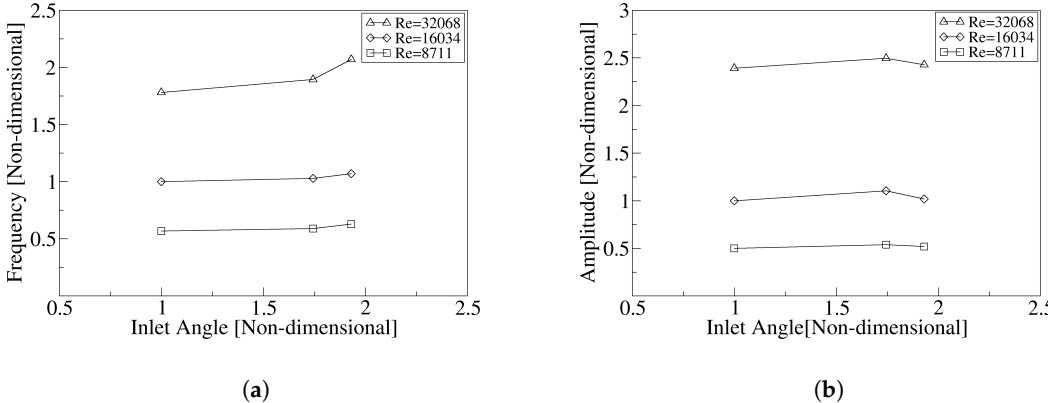

(**a**)  (**b**)

**Figure 13.** Fluidic oscillator output mass flow Frequency (**a**) and amplitude (**b**) as a function of the mixing chamber inlet angle and for three different Reynolds numbers, 8711, 16034 and 32068.

Figure 14 introduces the velocity field and the pressure magnitude inside the (FO) for the maximum and minimum (MC) inlet angles evaluated. As the inlet angle increases, there is less space on both sides of the jet in the (MC) for the Coanda effect to appear, yet the jet keeps oscillating. This supports the thesis presented in this paper and already outlined in [12,18], which established that in reality, what forces the jet to flip is the pressure term of the net momentum acting on the lateral sides of the jet entering the (MC). Regardless of the (MC) inlet inclination angle, the maximum stagnation pressure appearing at the (MC) converging walls is very similar, a peak to peak stagnation pressure increase of nearly 18% is observed when comparing the highest inlet angle evaluated with the lowest inlet angle case, compare Figure 14c with Figure 14d, see as well Figure 15c. As a result, the net momentum peak to peak amplitude acting onto the incoming jet lateral surfaces, suffers a small increase of 2.8%, although it is difficult to distinguish the different curves, this information is presented in Figure 15d.

From the observation of Figures 4a,b, 8a,b, 11a,b and 14a,b, it is noticed that in all (FC's) 90 degrees corners small vortices appear, indicating it exists fluid recirculation in all these points. To minimize fluid recirculation it would be desirable to round all 90 degrees corners, the expected effect would be, a small increase of actuator frequency, since in reality rounding the corners would facilitate the fluid to move back and forward along the feedback channels, pressure losses would as well decrease, This observation was previously done by [5].

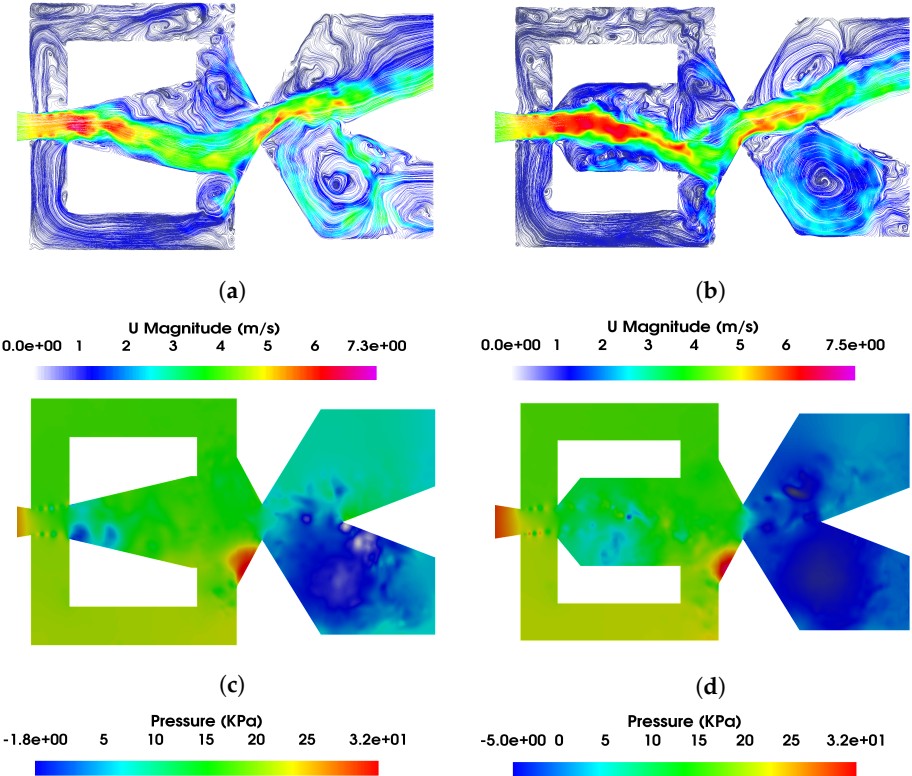

**Figure 14.** Fluidic oscillator internal velocity field (**a**,**b**) and pressure magnitude (**c**,**d**). Maximum inlet angle (**a**,**c**), minimum inlet angle (**b**,**d**). Reynolds number 16034.

Based on the results obtained in the present section, see Figure 15, it can be concluded that the effect of modifying the (MC) inlet angle, does not generate very relevant changes on any of the studied fluid flow parameters. In fact, the (MC) inlet angle seems to be particularly linked with the Coanda effect alternatively appearing on both sides of the mixing chamber. Yet, and based on the results obtained in the present set of simulations, the Coanda effect has a minor effect on the (FO) flow dynamic performance. It is important to realize, when observing Figure 15, that the minor (MC) inlet angle, corresponds to the baseline case, therefore, the other two angles studied are called medium and highest angles. The feedback channels mass flow peak to peak amplitude, suffered a decrease versus the baseline case of 14.5%, when the medium inlet angle was used, the (FC) peak to peak amplitude decreased a 16% when the inlet angle increase was of 93%, see Figure 15e. The reverse flow at the (FO) outlets suffered an initial increase as the inlet angle increased, and slightly reduced when the inlet angle reached its maximum value, see Figure 15a. The jet inclination angle inside the (MC), suffered a small decrease of 9% when comparing the minimum and maximum (MC) inlet angles evaluated, see Figure 15b. Notice that the jet inclination angle inside the (MC) is in reality delimited by the (MC) internal walls, as the inlet angle increases there is less space in the (MC) for the jet to fluctuate. From this particular study it is observed that, small stagnation pressure variations at the (MC) converging walls, generate clear (FO) outlet mass flow modifications, compare Figure 15a with Figure 15c.

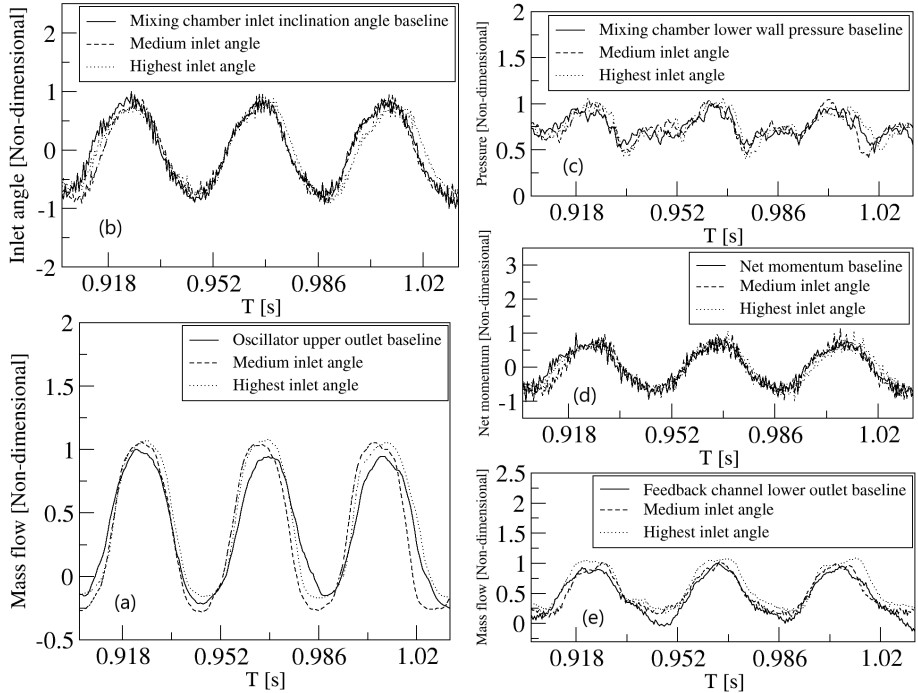

**Figure 15.** Dynamic effects of the (MC) inlet angle modification on the main flow parameters, Reynolds number 16034. Each graph presents three non-dimensional curves characterizing results from the baseline, the medium and highest inlet angle cases, and as a function of the dimensional time. In figure (**a**) the mass flow across the upper oscillator outlet is presented. Figure (**b**) introduces the temporal variation of the (MC) inlet inclination angle. Figures (**c**) presents the pressure at the (MC) lower inclined wall. The net momentum acting on the lateral sides of the main jet is presented in figure (**d**). Figure (**e**) characterizes the mass flow at the lower feedback channel outlet.

The corresponding videos presenting the velocity and pressure fields for the eight different cases evaluated, see Figures 4, 8, 11 and 14, are given in Supplementary Materials; a total of sixteen videos are introduced.

### 5.5. Relation Reynolds Frequency for All Dimensional Modifications Performed

After evaluating the fluidic oscillator output mass flow frequency and amplitude as a function of the different internal modifications and at several Reynolds numbers, one of the conclusions from the present paper is, that the conventional Reynolds-frequency linear behaviour for a given oscillator, can be expressed of a set of linear functions. Each line represents the operating conditions of the fluidic oscillator once a particular modification is undertaken, notice that in almost all cases, a linear relation is obtained, see Figure 16. For example, the increase of the inlet width, and regardless of the Reynolds number employed, generates output frequencies considerably higher than the baseline case. The (FO) outlet mass flow oscillating frequency, increases versus the baseline case one, by around 40% when the maximum inlet width is employed. The outlet frequency also increases when employing the lowest output angle or the lowest output width, although for these particular cases the increase is smaller than 7%. The rest of the internal modifications, generate frequencies slightly smaller than the baseline case ones, the trend is the same for all Reynolds numbers studied. It is also interesting to observe that, when the highest (MC) outlet angle is used, the expected Reynolds-frequency linearity disappears at high Reynolds numbers, the frequency for this case is smaller than what could be expected. The authors believe this particular reduction of frequency at Reynolds 32068, is due to the channel effect caused by the highest outlet angle, this particular angle directs the fluid from the (MC) to the oscillator outlets, minimizing the generation of a stagnation pressure point at the (MC) converging walls, in other words, just an small amount of the kinetic energy associated to the fluid is being transformed into stagnation

pressure. The result is, it requires a longer time to build the required momentum at the (FC) outlets for the jet to flip. Under these conditions, the fluid is directed to the lateral sides of the oscillator external chamber wedge and flows almost parallel to them, see Figure 11a.

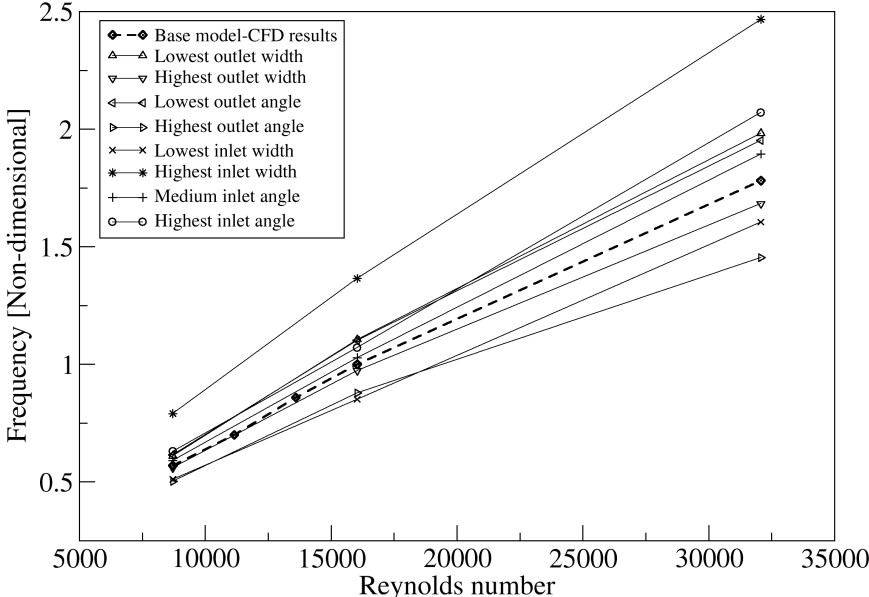

**Figure 16.** Relation Reynolds number versus mass flow output frequency, for all dimensional modifications studied.

It is at this point interesting to remember that, according to [3], the (FO) angled configuration, which is the one used in this study, looses its linearity at Reynolds 30000, this helps to explain why the curves presented in Figure 16, are not fully linear. As Reynolds number increases, the flow inside the (FO) goes from quasi-periodic to chaotic, being this the reason why linearity disappears. It is also interesting to recall, that according to [11], the fluidic oscillator internal performance is essentially the same for different outlet configurations, whether one or two outlets are considered.

*5.6. Stagnation Pressure at the (MC) Converging Walls and Net Momentum Acting on the (MC) Incoming Jet as a Function of the Reynolds Number*

In the present section, the net momentum acting onto the jet entering the mixing chamber will be evaluated and compared with the stagnation pressure variations at the (MC) converging walls. The momentum of the fluid acting on a given surface was defined as Equation (1). As both feedback channels, add momentum to the lateral sides of the main jet entering the (MC), Equation (1) will need to be applied to each (FC) outlet. In order to evaluate which is the temporal net momentum applied to the incoming jet, it will be required to know the instantaneous mass flow through both (FC) outlets as well as the temporal pressure at these two sections.

In [18] it was concluded the oscillators studied appeared to be pressure driven, the same conclusion was reached in [12], where it was demonstrated that the forces triggering the oscillation were mostly due to the pressure difference at the feedback channels outlets. In the present section the different forces acting on the jet lateral surfaces will be analyzed for each of the different geometry modifications evaluated and as a function of the Reynolds number.

The average net momentum applied to the jet entering the (MC), is for all Reynolds numbers and (FO) modifications, having a value close to zero, as observed for a Reynolds number of 16034

in Figures 5d, 9d, 12d and 15d. Regarding the evolution of the average net momentum applied to the jet at the (MC) inlet, it was observed that whenever the pressure at the (MC) is particularly high and the Reynolds number increases, the average net momentum applied to the jet decreases. This happens for the following cases, lowest outlet width, lowest outlet angle and for the baseline case. The equations presented in Table 4 characterize this evolution. On the other hand, whenever the pressure at the (MC) is particularly low, which happens for the highest outlet width, the lowest inlet width, the highest inlet width and the highest outlet angle, the average net momentum increases with the Reynolds number increase, Table 4 states this evolution. Nevertheless, and as a general trend it can be stated that, regardless of the (FO) modification performed, as the Reynolds number increases the average pressure in the (MC) also increases. The equations characterizing the evolution of the average non-dimensional stagnation pressure at the (MC) converging walls, and the average non-dimensional net momentum acting on the jet entering the (MC), are respectively presented as a function of the Reynolds number in Tables 3 and 4. These equations are valid for a range of Reynolds numbers between $8711 < \text{Re} < 32068$, and were obtained from the data presented in Figures 5, 9, 12 and 15, as well as from similar figures obtained at Reynolds numbers 8711 and 32068. Some geometry modifications, regardless of the Reynolds number, generate a decrease of the (MC) average pressure when compared to the one existing in the baseline case, these are, the highest and lowest inlet widths, the highest outlet angle and the highest outlet width. At Reynolds number 32068, the (MC) average pressure decrease versus the baseline one was respectively of 23.5%, 3.8%, 30% and 75%. On the other hand, the geometry modification generating a drastic increase of the (MC) average pressure, was the lowest outlet width, which increased the baseline pressure by almost 42 times at Reynolds 32068. From the equation presented in Table 3, characterizing the evolution of the average stagnation pressure at the (MC) converging walls, as a function of the Reynolds number and for the lowest outlet width case, it is clearly observed that the pressure increase is much higher under these conditions than for the rest of the cases studied. Nevertheless the rest of the equations presented in Table 3 show an increase of the average stagnation pressure at the (MC) converging walls with the Reynolds number increase. When employing the lowest outlet angle and for the same Reynolds number 32068, the average pressure increase was of 11.4%.

**Table 3.** Equations characterizing the evolution of non dimensional average stagnation pressure at the (MC) outlet converging walls, and as a function of the Reynolds number. These equations are valid in the range $8711 < \text{Re} < 32068$. The coefficient of determination was $(R)^2 = 1$ for all curves presented.

| Geometry Modification | Non-Dimensional Average Pressure at the (MC) Converging Walls, as a Function of the Reynolds Number |
|---|---|
| Baseline | $2.125E - 9 * Re^2 - 7.136E - 6 * Re + 0.5679$ |
| Highest inlet angle | $2.092E - 9 * Re^2 - 3.788E - 6 * Re + 0.5444$ |
| Lowest outlet angle | $2.054E - 9 * Re^2 - 6.890E - 7 * Re + 0.4840$ |
| Highest outlet angle | $1.935E - 9 * Re^2 - 9.367E - 7 * Re + 0.8536$ |
| Lowest outlet width | $1.144E - 7 * Re^2 - 4.102E - 4 * Re + 2.9272$ |
| Highest outlet width | $1.214E - 11 * Re^2 + 3.804E - 6 * Re + 0.4891$ |
| Lowest inlet width | $1.944E - 9 * Re^2 - 3.324E - 6 * Re + 0.5353$ |
| Highest inlet width | $1E - 9 * Re^2 + 1.655E - 5 * Re + 0.4195$ |

The analysis of the peak to peak amplitude of the stagnation pressure at the (MC) outlet converging walls, and the peak to peak net momentum amplitude acting on the lateral sides of the jet entering the (MC), provides a significant information on the flow dynamics inside the (MC). Tables 5 and 6, introduce the equations characterizing the evolution of these parameters as a function of the Reynolds number and for the different geometry modifications studied. The first thing to observe is that, the stagnation pressure and the net momentum amplitudes, increase as a function of the Reynolds number almost to the power 2. It is also interesting to observe that, in nearly all the cases studied, the exponent associated to the Reynolds number when considering the stagnation pressure peak to

peak amplitude, is smaller than the one associated to the net momentum amplitude for the same case. This is particularly relevant for the following cases, highest inlet width, highest inlet angle, lowest outlet angle and lowest outlet width. In any case, and regardless of the case studied, the equations from Tables 5 and 6, show a direct link between the peak to peak stagnation pressure amplitude and the net momentum amplitude, giving therefore strength to the thesis established in references [12,18] and in the present paper, regarding the origin of the forces driving the oscillation. To understand, for each of the cases studied, the origin of the net momentum driving the oscillation, the relation between the peak to peak net momentum term due to the static pressure acting on the (FC) outlets, was compared with the peak to peak net momentum term due to the feedback channels mass flow. This comparison given as the ratio between the static pressure divided by the (FC) mass flow term, is presented in Table 7. There are four geometry modifications, highest inlet width, highest inlet angle, lowest outlet angle and lowest outlet width, at which the pressure/mass flow momentum ratio is particularly small, indicating that under these conditions, the feedback channel mass flow plays a more relevant role regarding the net momentum applied to the jet. Notice that these four geometry modifications, are the same ones generating a particular increase in the exponent associated to the Reynolds number, observed when comparing Tables 5 and 6. The conclusion is that, for these particular geometry modifications, the (FC) mass flow plays a more relevant role, although small, on the net momentum driving the oscillation. Yet, regardless of the geometry modification and the Reynolds number studied, the oscillation of the jet in the (MC) is mostly driven by the pressure difference at the (FC) outlets. Finally, and based on the results presented in Table 7 it can be concluded that, the pressure/mass flow momentum ratio is highly dependent on the geometry modification, but it is not particularly affected by the Reynolds number.

**Table 4.** Equations characterizing the evolution of non-dimensional average net momentum acting on the jet entering the (MC), and as a function of the Reynolds number. These equations are valid in the range $8711 < \text{Re} < 32068$. The coefficient of determination was $(R)^2 = 1$ for all curves presented.

| Geometry Modification | Average Non-Dimensional Net Momentum Acting on the Jet Entering the (MC), as a Function of the Reynols Number |
|---|---|
| Baseline | $1E - 9 * Re^2 - 8.375E - 5 * Re + 0.5899$ |
| Highest inlet angle | $1.243E - 8 * Re^2 - 1.532E - 4 * Re + 0.2218$ |
| Lowest outlet angle | $3.374E - 9 * Re^2 - 1.144E - 4 * Re + 0.9911$ |
| Highest outlet angle | $7.508E - 11 * Re^2 - 4.282E - 5 * Re + 0.3290$ |
| Lowest outlet width | $-3.842E - 8 * Re^2 + 8.762E - 4 * Re - 4.2989$ |
| Highest outlet width | $4.833E - 9 * Re^2 - 6.425E - 5 * Re - 0.3056$ |
| Lowest inlet width | $2.189E - 9 * Re^2 - 2.47378E - 5 * Re + 0.2035$ |
| Highest inlet width | $9.226E - 9 * Re^2 - 3.57369E - 4 * Re + 2.3774$ |

**Table 5.** Equations characterizing the evolution of non-dimensional peak to peak stagnation pressure amplitude at the (MC) outlet converging walls, and as a function of the Reynolds number. These equations are valid in the range $8711 < \text{Re} < 32068$.

| Geometry Modification | Non-Dimensional Peak to Peak Pressure Amplitude at the (MC) Converging Walls | Coefficient of Determination $(R)^2$ |
|---|---|---|
| Baseline | $4.597E - 9 * Re^{1.986}$ | 0.999 |
| Highest inlet angle | $7.493E - 9 * Re^{1.956}$ | 0.999 |
| Lowest outlet angle | $1.939E - 8 * Re^{1.851}$ | 0.998 |
| Highest outlet angle | $1.237E - 9 * Re^{2.066}$ | 0.997 |
| Lowest outlet width | $2.126E - 8 * Re^{1.865}$ | 0.997 |
| Highest outlet width | $6.077E - 9 * Re^{1.937}$ | 0.999 |
| Lowest inlet width | $2.820E - 9 * Re^{2.020}$ | 0.993 |
| Highest inlet width | $1.872E - 8 * Re^{1.839}$ | 0.993 |

**Table 6.** Equations characterizing the evolution of non-dimensional peak to peak net momentum amplitude at the (MC) incoming jet, and as a function of the Reynolds number. These equations are valid in the range 8711 < Re < 32068.

| Geometry Modification | Non-Dimensional Peak to Peak Net Momentum Amplitude at the (MC) Incoming Jet | Coefficient of Determination $(R)^2$ |
|---|---|---|
| Baseline | $4.652E-9*Re^{1.983}$ | 0.999 |
| Highest inlet angle | $2.706E-9*Re^{2.044}$ | 0.999 |
| Lowest outlet angle | $5.633E-9*Re^{1.971}$ | 0.999 |
| Highest outlet angle | $1E-9*Re^{1.929}$ | 0.989 |
| Lowest outlet width | $4.298E-9*Re^{2.002}$ | 0.999 |
| Highest outlet width | $2.908E-9*Re^{2.004}$ | 0.997 |
| Lowest inlet width | $2E-9*Re^{2.006}$ | 0.997 |
| Highest inlet width | $1.864E-9*Re^{2.038}$ | 0.998 |

**Table 7.** Evaluation of the peak to peak net momentum amplitude at the (MC) incoming jet due to the pressure term, divided by the net momentum amplitude due to the (FC) mass flow term, and for the three Reynolds numbers studied.

| Geometry Modification | Reynolds Number 8711 | Reynolds Number 16034 | Reynolds Number 32068 |
|---|---|---|---|
| Baseline | 11.96 | 12.58 | 9.88 |
| Highest inlet angle | 8.72 | 9.38 | 8.1 |
| Lowest outlet angle | 9.73 | 10.71 | 9.25 |
| Highest outlet angle | 16.86 | 17.14 | 14.97 |
| Lowest outlet width | 10.63 | 10.77 | 9.32 |
| Highest outlet width | 12.93 | 13.29 | 10.88 |
| Lowest inlet width | 13.61 | 12.38 | 14.52 |
| Highest inlet width | 3.78 | 3.74 | 4.12 |

Regarding the inlet width variations, from the results obtained it is observed that, the stagnation pressure and the net momentum peak to peak amplitudes, suffer a minor increase as the inlet width increases. When comparing the minimum and maximum inlet widths studied, at Reynolds number 16034, the peak to peak stagnation pressure and momentum increase, were respectively of 17% and 31% versus their minimum values. When considering the outlet width effects on the (MC) converging walls peak to peak stagnation pressure amplitude, it was observed that as the outlet width decreases, the peak to peak stagnation pressure amplitude keeps increasing, also the net momentum amplitude acting on the lateral surfaces of the (MC) incoming jet, increases as the outlet width decreases. When comparing the minimum and maximum outlet widths studied, the increase of the peak to peak stagnation pressure and momentum, at Re = 16034, was respectively of 39% and 36%. The same trend is observed under all Reynolds numbers studied. As Reynolds number increases, the variation in percentage for both parameters decreases.

When evaluating the (MC) outlet angle and for all the cases studied, the minimum peak to peak (MC) converging walls stagnation pressure, was obtained when the highest outlet angle was employed. As the outlet angle decreased, the peak to peak stagnation pressure at the (MC) outlet converging walls, as well as the peak to peak momentum amplitude at the (FC) outlets, kept increasing. The increase versus the minimum value was respectively of 68% and 38% at Re = 16034. As Reynolds number grows, the variation in percentage reduces. When using the highest outlet angle, the peak to peak net momentum amplitude associated was higher than for the case where the lowest inlet width was used. The decrease of the inlet width, appears to particularly reduce the net momentum applied to the (MC) incoming jet. The mass flow along the (FC's) is effectively controlled by the inlet width. Regardless of the Reynolds number, as the inlet angle increases, the stagnation pressure peak to peak amplitude at the (MC) converging walls, also increases. The increase is more relevant at high Reynolds

numbers. It is interesting to see that the peak to peak amplitude of the net momentum acting on the lateral sides of the jet entering the (MC), initially increases with the inlet angle increase, but as the inlet angle reaches its maximum value, the momentum amplitude slightly decreases. At the highest inlet angle, the stagnation pressure waves generated at the (MC) converging walls, are not efficiently being transferred to the (FC's) outlets. In fact, the pressure at both (FC's) outlets, is very much the same under these conditions. Regardless of the (FO) geometry modification performed, the same trend on peak to peak pressure and momentum, appeared at all Reynolds numbers evaluated, therefore, for a given (FO) modification, the same physical phenomenon is driving the oscillations at all Reynolds numbers.

In order to properly understand the effect of the inlet width variation on the net momentum acting onto the (MC) incoming jet, Figure 17 is introduced. The first point to realize is that, for the lowest inlet width, the noise associated to the pressure wave generated at the stagnation points appearing alternatively at the (FC) outlets, see Figures 4d and 5d, is much higher than in the rest of the cases studied. For this particular case, see Figure 5e, there is a large amount of mass flow traveling backwards along the feedback channels. The authors believe, the weak pressure waves originated at the (MC) outlet converging walls, which travel along the (FC) and also inside the (MC), are being disrupted by the particularly high mass flow moving along the (FC) and in opposite direction to the pressure waves, the result is a highly noisy pressure fluctuation at the (FC) outlets. In reality, and for this particular case, pressure waves are being generated at the same time on the (MC) outlet converging walls and at the (FC's) outlet internal vertical walls, these pressure waves collide inside the (FC's) enhancing the noise spectrum. Notice as well from Figure 17 that, the noise generated reduces to a minimum whenever the maximum inlet width is being employed. For this case, the mass flow traveling along the (FC's), always goes from the (FC's) inlets to the outlets, see Figure 5e, and pressure waves are not generated alternatively at the (FC's) outlet internal vertical walls. The direction of the (FC's) mass flow, always coincides with the traveling direction of the pressure waves, which are generated at the (MC) outlet converging walls.

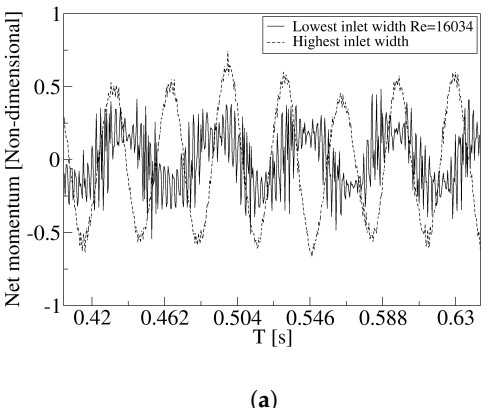
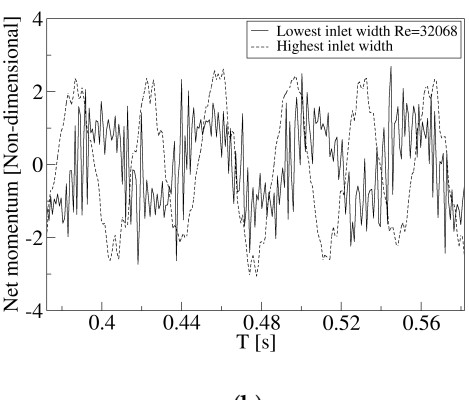

(**a**)                              (**b**)

**Figure 17.** Net momentum acting on the fluidic oscillator inlet jet and for two different mixing chamber inlet widths (the lowest and the highest), two different Reynolds numbers, 16034 Figure (**a**) and 32068 Figure (**b**), were considered.

## 6. Conclusions

A careful 3D-CFD evaluation of a fluidic oscillator under turbulent conditions has been performed. The study allows identifying, which dimensional parameters are more relevant regarding the modification of the fluidic actuator frequency and output amplitude. When modifying the (MC) inlet width, a threshold in both directions was observed at which fluidic oscillator was simply not oscillating. By increasing the (MC) outlet width or the (MC) outlet angle, the (FO) output frequency and amplitude decreased. The maximum (FO) outlet mass flow amplitude, was always obtained

whenever reverse flow at the (FO) outlets existed, the higher the reverse flow value, the higher the (FO) outlet mass flow amplitude.

The pressure term of the net momentum acting onto the lateral sides of the mixing chamber incoming jet, directs under all conditions studied, the oscillation of the jet inside the (MC) and therefore the oscillation at the (FO) outlets. The actuator is pressure driven. The net momentum oscillation is mostly due to the stagnation pressure fluctuation occurring at the (MC) converging surfaces, the net momentum due to the mass flow flowing along the feedback channels was observed to be negligible in all cases studied, the amplitude of the net momentum oscillation is directly linked with the maximum and minimum values of the stagnation pressure appearing at the (MC) outlet converging surfaces.

Low inlet widths, have associated a considerable reverse flow along the (FC's), reverse flow also appears at the (FO) outlets, therefore the (FO) mass flow amplitude is higher than the one existing at high inlet widths. The net momentum required to flip the jet over in the (MC), is particularly low at small inlet widths. The modification of the inlet width, drastically affects the magnitude and direction of the (FC) mass flow. At very small inlet widths, pressure waves are generated on both ends of the (FC's), generating high levels of noise which affects the net momentum acting onto the (MC) incoming jet. The variation of the (MC) outlet width, affects mostly the (FO) output amplitude, from all cases studied, the highest outlet width generates the smallest stagnation pressure peak to peak amplitude, the smallest peak to peak net momentum, the smallest peak to peak (FO) output mass flow, the smallest peak to peak inlet angle amplitude and the smallest (FC) mass flow amplitude.

As the (MC) outlet angle increases, the average stagnation pressure at the (MC) converging walls, as well as its peak to peak amplitude decreases, a reduction of all peak to peak parameters is observed. In general it can be said that, the trend defined by the (MC) outlet converging walls stagnation pressure amplitude, is followed by the amplitude of the rest of the variables, the (MC) inlet angle, the net momentum applied to the (MC) incoming jet, the (FO) output mass flow and the (FC) mass flow.High frequencies are linked with high stagnation pressure values. Oscillator mass flow amplitude directly depends on the reverse flow appearing at the (FO) outlets. Reverse flow is particularly high at lowest outlet widths.

**Supplementary Materials:** The following are available online at https://zenodo.org/record/3725490. A set of sixteen videos introducing the flow and pressure distribution at Reynolds number 16,034, and characterizing the different extreme cases initially described in Figure 4 for the maximum and minimum inlet width, Figure 8 for the maximum and minimum outlet width, Figure 11 characterizing the maximum and minimum outlet angle and Figure 14 for the maximum and minimum inlet angle, are also provided.

**Author Contributions:** Conceptualization, M.B. and J.M.B.; methodology, M.B. and J.M.B.; software, M.B.; validation, M.B.; formal analysis, M.B. and J.M.B.; investigation, M.B. and J.M.B.; data curation, M.B.; writing—original draft preparation, M.B. and J.M.B.; writing—review and editing, M.B. and J.M.B.; visualization, M.B. and J.M.B.; supervision, J.M.B.; project administration, J.M.B.; funding acquisition, J.M.B. All authors have read and agreed to the published version of the manuscript.

**Funding:** This work was supported by the Spanish and Catalan Governments under grants FIS2016-77849-R and 2017-SGR-00785, respectively. Part of the computations were done in the Barcelona Supercomputing Center under grants FI-2016-3-0038, FI-2017-2-0020, FI-2017-3-0009, FI-2018-3-0036 and FI-2019-1-0017.

**Conflicts of Interest:** The authors declare no conflicts of interest.

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
