# Peer review of "Fluidic Oscillators, the Effect of Some Design Modifications"

_applsci, doi:10.3390/app10062105_

Round 1

Reviewer 1 Report

Review of “Fluidic Oscillators, the effect of some design Modifications” by Masoud Baghaei and Josep M. Bergada

The paper presents a CFD study of the 3D flow inside fluidic oscillators. The authors examine the effect of specific design modifications on the flow pattern inside the device, and examine scalings of the momentum “input” and frequency/amplitude of the “output” generated by the devices. The study is well-planned and well-written, with results that -while not groundbreaking- is interesting and noteworthy. The paper should be published once the authors address some of the minor issues related to their study, and one “major” issue detailed below.

Turbulence modeling may be the most important aspect in the present CFD. The tortuous passages in the actuator devices will be tough to capture well in CFD, due to the separations, flow reversals, recirculations, etc. that may happen in the flow. The authors must provide a little more information regarding verification & validation of their code. Currently they provide one reference (4) to work by others. They must explain what steps they took to make their CFD valid, including mesh independence/grid convergence studies, comparisons with experiment/previous CFD, choice of CFD parameters, etc. Currently their manuscript reports on the choices that they made, but provides little in the way of explanation and justification of these choices. This is one aspect of the paper that must be addressed, and it constitutes the only major change that needs to be presented before publication.

Other comments, more secondary in nature:

The authors put a lot of trust in the numerical accuracy of their CFD, up to and including the definition of non-dimensional numbers: when they report their numbers as “8711, 11152, 13593, 16034, 32068” they are assuming 5/6 significant digits in their calculations. These are probably best presented as 8700, 11200, 13600, 16000, 32000, considering that values of viscosity, temperature, etc. are all accurate to within a few percentage points anyway.

Similarly with Tables 3-6: data presentation assumes 7/8 significant digits for things like linear-squares fits, and similar correlations based, for the most part, on 3 different data points. Do the authors really expect such accuracy in their code, and do they expect the physical phenomena to scale with Reynolds number to 1.9985, say, or 2? Put another way, the authors should attempt to interpret their numerical fits, and try to find a physical explanation for a quadratic scaling with Re number. If this is the case, perhaps the authors could consider eliminating the somewhat repetitive tables, and present data as in figure 16, in log-log axes, with a power law (straight line) fit corresponding to Re-squared. This may be a more effective method of quantitative presentation.

Is it really the Reynolds number that is important here, or is it simply an effect of velocity (squared)? Could it not be something related to an inviscid equation (perhaps Bernoulli) where the viscosity plays no role? The authors use the term “stagnation pressure” throughout, and, considering that theirs is an incompressible computation, the term may end up confusing the reader. Strictly speaking in a low-speed incompressible fluid stagnation and static pressures should be identical; the authors may want to verify this from their CFD results and, if it proves to be the case, eliminate the term “stagnation”.

Author Response

Please see the file attached.

Reviewer 2 Report

A fluidic oscillator was numerically simulated in order to examine the effects of design parameters on its performance such as output frequency and amplitude. The software OpenFOAM employing PISO (Pressure Implicit with Splitting Operators) was utilized for the simulation of 3-dimensional, incompressible, isothermal, and turbulent water flow.

Based on the boundary conditions given by the authors, this paper succeeds in scrutinizing the effects of oscillator's dimensions and their roles playing through a structural variation of the flow in the oscillator.

(1) w.r.t. the theoretical (mathematical) model

Although the OpenFOAM was used for the simulation and the incompressible Navier-Stokes equation is not that complex, a thorough description for the governing set of equations along with the relevant transport properties and turbulent constants used in the computation shall be presented not only for the appropriate verification by others but also for the dissemination of fruitful information to the paper readers.

(2) w.r.t. the computational domain and boundary conditions

Fig. 4, 8, 11, & 14 reveal that the internal flow structure including the FO outlet is an elliptic type (or two-way coordinate system) flow.

Is it justifiable to endow the FO outlet with Neumann boundary conditions (whatever the dependent variable is)?

Otherwise, the external region downstream of the FO outlets shall be necessarily counted in the computation. 

(3) Typo-errors

  • Line # 109: finite volumes -> finite volume method
  • Line # 217: figure 5a -> figure 4a
  • Line # 282: studding -> studying
  • Line # 397: 4200 Pascals -> 4200 Pa
  • Line # 612: flow tern -> flow term

Author Response

Please see the file attached.

Round 2

Reviewer 1 Report

The authors addressed all issues raised during review and the paper can be published in its present form